# Nanoscale Functional Additives Application in the Low Temperature Greases

**DOI:** 10.3390/polym13213749

**Published:** 2021-10-29

**Authors:** Sergey A. Shuvalov, Yaroslav V. Porfiryev, Dmitry S. Kolybelsky, Vladimir A. Zaychenko, Pavel S. Popov, Pavel A. Gushchin, Alexandr S. Mishurov, Daria A. Petrova, Vladimir A. Vinokurov

**Affiliations:** 1NC Rosneft-MP Nefteprodukt, PJSC, 40 Shosse Entuziastov, 105118 Moscow, Russia; shuvalov_sa@mail.ru (S.A.S.); mz_research@mail.ru (Y.V.P.); kolubelskyds@yandex.ru (D.S.K.); popov_ps@list.ru (P.S.P.); 2Department of Physical and Colloid Chemistry, Gubkin University, 65 Leninsky Prospekt, 119991 Moscow, Russia; zaychenko_v@mail.ru (V.A.Z.); guschin.p@mail.ru (P.A.G.); mishurov99@list.ru (A.S.M.); petrova.msu@gmail.com (D.A.P.); 3Skolkovo Institute of Science and Technology, 121205 Moscow, Russia; 4Faculty of Chemistry, Moscow State University, GSP-1, Leninskie Gory, 119991 Moscow, Russia

**Keywords:** halloysite, complex lithium grease, polyurea grease, polymer grease, nanoscale particles, montmorillonite, calcium carbonate, calcium borate, silicon dioxide

## Abstract

Due to the fact that the application of AW and EP additives in low-temperature greases may lead to worse high-temperature and anti-corrosion characteristics as well as additional burden on the environment due to the content of aggressive components, in this paper, the possibility of replacing these additives with NFA, which do not have these disadvantages, was investigated. The analysis of nanosized particles being used as functional additives in greases was carried out. The morphology of the following nanoparticles was studied: montmorillonite K 10, silica, calcium car-bonate and borate, halloysite, and molybdenum disulfide incorporated in halloysite tubes. The effect of nanostructured components on the physicochemical characteristics and anti-wear and anti-scuffing properties of complex lithium, polyurea, and polymer greases were studied. Maximal improvement of anti-wear and anti-scuffing characteristics of cLi-greases was reached when using silica and calcium borate. Maximal improvement of anti-scuffing properties of PU-lubricant was reached when using calcium carbonate and the two-component NFA based on halloysite, for anti-wear properties when adding silicon dioxide and halloysite. When the concentrations of silicon dioxide and calcium carbonate was increased from 1 to 3 wt.%, there was a decrease in yield stress of the structural frame of the PU-lubricant and its colloidal stability was worse. The increase of the concentration of calcium carbonate and borate nanoparticles in the studied range led to a significant improvement of the anti-wear and anti-scuffing characteristics of the PU grease, respectively. The greases properties’ dependence from the nanostructured functional additives’ introduction method and their concentration were investigated. Nanoparticles were added into the test lubricants before and after the thermo-mechanical dispersion stage. The addition of silicon dioxide and calcium carbonate NFA after the heat treatment stage led to worsening of the characteristics of the plastic material, and the increase of their concentration from 1 to 3 wt.% formed a harder structure of Li-grease. On the contrary, the addition of calcium borate NFA is recommended after the thermomechanical dispersion. The choice of nanoparticles and the method of their addition to the lubricants of various types was carried out according to the results of the previous stage of the research. Along with the analysis of the physicochemical characteristics and anti-wear and anti-scuffing properties of the lubricants, the structure of the dispersion phase of nanomodified lubricants were studied.

## 1. Introduction

Low-temperature greases (LTGs) must have excellent lubricating properties and be able to ensure reliable operation of the friction units of mobile and stationary equipment under high loads. Currently, for lubricating properties’ evaluation, producers of low-temperature greases use wear scar and welding load indicators characterizing anti-wear and anti-scuffing properties, respectively. In order to improve the performance properties of LTGs, anti-wear (AW) and extreme pressure (EP) additives are widely used, which, when thermally decomposed, lead to the formation of sulfur–phosphorus components, creating a multilayer anti-wear film when reacting with the structural materials in the contact zone [1]. To prevent serious wear on surfaces, the concentration of such functional additives is up to 5% of weight.

The application of AW and EP additives in low-temperature greases based on low-viscosity mineral base oil along with improvement of the anti-wear and anti-scuffing properties may lead to worse high-temperature and anti-corrosion characteristics as well as to additional burden on the environment due to the content of aggressive components [2,3].

A promising direction for improving the anti-wear and anti-scuffing properties is the use of nanostructured functional additives (NFA). It is known that nanosized carbon materials like nanotubes [4,5,6], graphene [7,8], nanodiamonds [9,10,11], metal nanoparticles, and its oxides [12,13,14,15,16,17] are widely used as fillers for greases. Some metal containing and boron compounds can also be used [18,19,20,21,22,23,24,25].

Among the advantages of nanoscale additives are the possibility to fill the roughnesses in the contact point, the thermal stability [26], and the high reaction rate with metal surfaces without an induction period inherent to the traditional AW and EP additives [27].

The positive effect of nanoparticles for anti-wear and anti-scuffing characteristics include the following [12,28]:Spherical nanoparticles slide between the friction surfaces and separate them, acting as micro-bearings.Nanoparticles can tribochemically react with materials of friction surfaces and form a protective chemisorption film on them.Microcracks and micronods on the damaged surface can be filled with nanoparticles, and as a result flat friction surface is restored.

The size, shape, nature, and concentration of the nanoparticles are very important factors affecting the performance of the greasing material. Much attention is paid to halloysite as a suitable material for surface modification nanotubes [29,30]. The addition of nanoparticles into the lubricating compositions is a difficult task, because they have high surface energy, because of which they tend to aggregate and coagulate after being introduced into most liquids. With changes in temperature or pressure and weak dispersion, these trends intensify [31]. Thus, depending on the method of addition to the lubricant composition (before or after thermomechanical dispersion), NFAs can have a significant effect on the process of structure formation of the lubricant and its physicochemical and operational characteristics.

The aim of this work is to study the dependence of the main physicochemical and anti-wear and anti-scuffing characteristics of low-temperature greases with various thickeners on the type, concentration, and method of introduction of nanostructured functional additives.

## 2. Materials and Methods 

### 2.1. Synthesis and Study of the Nanostructured Functional Additives

The following commercial nanostructured functional additives (NFA) were used in this study: montmorillonite K-10 (Sigma-Aldrich, St. Louis, MI, USA), silicon dioxide Kovelos 35/01T (Ecokremniy, Russia), and halloysite (Sigma-Aldrich, St. Louis, MI, USA).

Calcium borate nanoparticles were synthesized according to the procedure described in [32]. The synthesis of calcium carbonate NFAs was carried out according to the procedure [18]. Molybdenum disulfide nanoparticles incorporated into halloysite tubes were obtained according to the procedure described in [33].

For each NFA, the specific surface area was determined, and its morphology was studied.

Particle surface area was estimated by the Gemini VII 2390 Surface Area Analyzer’s physical adsorption study apparatus (Micromeritics Instrument Corporation, Norcross, GA, USA) (Figure 1) at the temperature of 77 K. Before measurements, the samples were degassed at a temperature of 300 °C for 4 h. The specific surface was calculated by the Brunauer–Emmett–Teller method based on adsorption data in the relative pressure range (P/P0) = 0.04–0.20. Pore volume and diameter were determined by the adsorption branch of isotherms using the Barrett–Joyner–Halend and Langmuir models. The specific pore volume was calculated by the amount of nitrogen adsorbed at the relative pressure P/P0 = 0.95.

The study of the morphology and the microstructure of the used nanoparticles was carried out by transmission electron microscope Jeol JEM-2100 (JEOL, Japan, Tokyo). For the transmission electron microscopy, 1 mg of nanoparticle powder was taken from the initial sample and diluted with distilled water in the ratio of 1:500.

The characteristics of the NFAs are given in the Table 1.

### 2.2. Preparation of Low-Temperature Plastic Greases

LTGs were prepared on the low-setting base oil S-9 produced by PJSC Bashneft. The characteristics of the dispersion medium are shown in Table 2.

The following compounds were used as the thickener:− Lithium complex soap based on 12-hydroxystearic and sebacic acids; − Polyurea based on polyisocyanate, octadecylamine, and aniline; − The mixture of polypropylene with different molecular weights. 

#### 2.2.1. Preparation of LTG Samples Based on the Complex Lithium Thickener 

A reactor with a stirrer, equipped with a heating jacket with a high temperature coolant, was loaded with S-9 (Bashneft, Ufa, Russia) (70% of total amount) and heated under constant mixing to the temperature of 90 ± 5 °C. Further, the calculated amount of thickener components was loaded, such as 12-hydroxystearic acid (Amee castor & derivatives LTD, Chandisar, India), sebacic acid (Hengshui Jinghua Chemical Co, Hebei, China), and 20% water solution of lithium hydroxide (Omkirmet-plus, Ekaterinburg, Russia), and saponification was carried out. Then, the reaction mixture was heated up to 125–130 °C to remove moisture, and then the soap oil concentrate was added with a pre-dispersed base oil NFA and exposed to the thermomechanical dispersion with constant mixing and a rise in the temperature up to 220 ± 2 °C and further cooling (Case 1) or NFA injected into the grease melt at the end of the thermomechanical dispersion stage, after which the crystallization of the fibers and the formation of the modified structural frame of the lubricant occurred (Case 2).

#### 2.2.2. Preparation of LTG Samples Based on the Polyurea Thickener 

In the reactor with the stirring device, the mixture of calculated amounts of octadecylamine (Salium Oleochemicals GmbH, Dessau-Roslau, Germany) and aniline (Spectrhim, Moscow, Russia) with the base oil S-9 (70% of total) was prepared at the temperature of 85 ± 10 °C. Then, small portions of the calculated amount of the previously prepared polyisocyanate solution in oil S-9 were loaded, after which the resulting mixture was heated under constant stirring up to the temperature of 130 ± 10 °C, held at this temperature for 30 min, and cooled. The addition of NFA was carried out by preliminary dispersion together with the mixture of amines (Case 1), or after the thermomechanical treatment stage in the form of the suspension of 10% of the total amount of base oil (Case 2).

#### 2.2.3. Preparation of the LTG Based on the Polymer Thickener 

In the reactor with the stirrer, the mixture of calculated amounts of polypropylene (LLC “NPP” Neftekhimiya”, Moscow, Russia) with different molecular masses and base oil was made. Next, the obtained mixture was heated under constant stirring up to the temperature of 200 ± 10 °C (thermomechanical dispersing stage), then rapid cooling of the melt was conducted. The NFA was introduced to the polymeric lubricant by pre-dispersion in the total amount of base oil (Case 1), or as the suspension in 10% of the total amount of oil directly before cooling of the lubricant melt (Case 2).

After cooling, lubricant created under the procedures in Section 2.2.1, Section 2.2.2 and Section 2.2.3 was mechanically treated with a colloid mill to obtain a homogeneous thixotropic structure.

### 2.3. Analysis of Low-Temperature Greases

To study the physicochemical and anti-wear and anti-scuffing characteristics of the LTGs, the following standard methods were used:− ASTM D 566 “Standard Test Method for Dropping Point of Lubricating Grease”; − GOST 7142-74 “Greases. Methods of determining colloidal stability”; − GOST 7143-73 “Greases. Method of determining the ultimate strength and thermohardening” (method B);− GOST 7163-84 “Mineral oils. Method of determining the viscosity by automatic capillary viscometer”;− GOST 9566-74 “Greases. Method of determining of evaporation “for 1 h at a temperature of 120°”;− GOST 9490-75: “Liquid lubricating and plastic materials: Method of test for lubricating characteristics on a four-ball machine.”

## 3. Results and Discussion 

### 3.1. The Study of the Morphology and Microstructure of the Used Nanoparticles

The nanostructured montmorillonite K-10 had a layered structure formed by needle-shaped particles with the length of 100–300 nm and a thickness of less than 5 nm (Figure 1). The nanosized particles of silicon dioxide had a spherical shape with an average diameter of about 20 nm. The particles of calcium borate also had a spherical shape with the average diameter of 20–40 nm. The nanostructured calcium carbonate was represented by cubic particles with the average side length of 40–50 nm. The particles of halloysite had a cylindrical shape with a length of 200–700 nm and an average diameter of 40 nm. The nanosized particles of molybdenum disulfide incorporated on the surface of halloysite tubes formed volumetric systems with the inhomogeneous surface.

### 3.2. Investigation of the NFA Influence on the Characteristics of the Lithium Complex, Polyurea, and Plastic Lubricants

At this stage, the influence of the type and method of addition of 1 wt.% nanoscale additives on the physicochemical and anti-wear and anti-scuffing properties of complex lithium (cLi), polyurea (PU), and the polymer (PP) lubricants was investigated. The results of the study are shown in Appendix A, respectively.

#### 3.2.1. The Effect of Montmorillonite K-10

The addition of this NFA into the Li-lubricant led to a slight decrease in yield stress and dropping point that apparently indicates the montmorillonite nanoparticles’ negative influence on the formation of the complex lithium soap, regardless of the mode of NFA addition. In both cases, lubricant anti-scuffing properties were improved slightly, but the anti-wear characteristics worsened remarkably, which implies the abrasiveness of montmorillonite nanoparticles under low loads and, consequently, the low efficiency of their use as a lubricity additive.

The addition of NFA before the heat treatment during the preparation of PU grease led to the increase in the yield stress, dropping point, and effective viscosity. The improvement of colloidal stability was observed regardless of the mode of NFA addition. The dependence of the anti-wear and anti-scuffing properties of the PU grease on the montmorillonite K-10 addition was similar to that of the cLi-grease.

The addition of the nanoscale additive in the PP lubricant led to the increase in the yield stress and effective viscosity. A slight improvement in colloidal stability was observed. The anti-wear and anti-scuffing characteristics of the PP grease remained almost unchanged. The method of NFA addition did not have any effect on the characteristics of the PP lubricant.

#### 3.2.2. The Silicon Dioxide Impact

The addition of silica (Kovelos 35/01T) in the cLi-grease after the heat treatment reduced the dropping point, which was probably due to the negative impact of NFA polar molecules on the formation of the complex lithium soap. The absence of a similar effect when the NFA was introduced before the heat treatment is explained by the screening of the silicon dioxide molecules’ active centers before the start of the complexation process. Regardless of the method of addition, the anti-scuffing characteristics were improved when nanosized particles of SiO_2_ were added, while anti-wear properties stayed almost the same.

The NFA addition to the PU-greases led to an increase of the yield stress, dropping point, and effective viscosity, but also to the improvement of colloidal stability and anti-wear and anti-scuffing characteristics.

The addition of aerosil nanoparticles into the PP-grease before heat treatment led to the increase of the yield stress and effective viscosity, as well as to the improvement of colloidal stability. NFA addition after the thermomechanical dispersing stage had a significant impact on the PP-grease properties. The anti-wear and anti-scuffing properties of PP-grease after its modification with silicon dioxide remained the same.

#### 3.2.3. The Calcium Carbonate Impact

The method of calcium carbonate nanoparticles addition in the cLi-grease composition determined the difference in the values of such quality indicators as yield stress and dropping point. The mechanism of influence of this NFA was likely similar to that of silicon dioxide. In this case, the degree of the nanostructured calcium carbonate impact on colloidal stability, the effective viscosity, and anti-wear and anti-scuffing characteristics of cLi-grease was negligible.

The inclusion of calcium nanocarbonates in the structure of PU-grease led to the increase of yield stress, dropping point, and effective viscosity, and to significant improvement in colloidal stability, and this effect was more strongly manifested when the NFA was introduced prior to the heat treatment stage. Of particular note was the almost doubled increase in the welding load of the modified PU-grease, which confirms the effectiveness of NFAs as the anti-scuffing component of greases based on the polyurea thickener.

The result of the modification of PP-grease with calcium carbonate nanoparticles was the reduction of the yield stress with simultaneous improvement of the colloidal stability. At the same time, the more significant improvement of anti-scuffing properties with the NFA addition after the heat treatment stage should be marked.

#### 3.2.4. The Calcium Borate Impact

The most noticeable impact of the NFA addition method to cLi-grease characteristics was noticed when the calcium borate was used. Its addition before the saponification reaction had a negative effect on the structure formation of the lubricant, which was expressed by a drastic decrease in the yield stress, dropping point, and colloidal stability. Probably, calcium borate, unlike silicon dioxide and calcium carbonate, due to its partial solubility in water, interacts with the components of the thickener with the formation of byproducts (calcium 12-hydroxystearate, calcium sebacate, etc.) that impede the normal saponification flow. When calcium borate was added after the heat treatment stage, the NFA particles participated in the formation of the thickener structure with the formation of systems similar to those of the three-component complex lithium soap described in [34].

The degree of calcium borate’s influence on the physicochemical characteristics of PU-greases was also dependent on the stage of NFA addition; when it is introduced before the final heat treatment stage one could observe the increase of yield stress, dropping point, and grease viscosity along with the colloidal stability improvement, while the anti-wear and anti-scuffing characteristics were not the subject of influence of the NFA addition method.

The inclusion of calcium borate to the PP-lubricant composition, regardless of the stage, led to colloidal stability improvement, whereas the other indicator values remained almost unchanged.

#### 3.2.5. Effect of Halloysite and Molybdenum Disulfide Incorporated into Halloysite Tubes

The addition of halloysite and molybdenum disulfide incorporated into halloysite tubes into greases with various thickeners has similar effect on their physicochemical and anti-wear and anti-scuffing properties.

In cLi-greases, when the NFA is introduced after the heat treatment stage the decrease in dropping point and worsening of the colloidal stability is observed. There is slight improvement in anti-scuffing characteristics when both NFAs are added regardless the stage of their addition.

The results of NFA addition to the PU-grease before the maximal preparation temperature is reached are yield stress increasing, improved colloidal stability, dropping point and effective viscosity rise, whereas, when the NFA is introduced after the heat treatment stage, the mentioned indicators’ values hardly change in comparison with the base grease. Modification of PU-grease with two-component NFA leads to significant improvement of anti-scuffing characteristics, while, when halloysite nanotubes are added, a wear scar diameter decreases.

The addition of these NFAs into PP-grease prior to the heat treatment stage increases the yield stress and improves colloidal stability. In all samples, slight decrease in the effective viscosity is observed. When modifying the PP-grease with two-component NFA, drastic improvement of anti-wear properties is observed and slight improvement of anti-scuffing characteristics, moreover the NFA addition after the thermomechanical dispersing gives greater positive effect.

Thus, as a result of the study of the influence of the type and method of NFA addition on characteristics of the greases based on various thickeners, several conclusions can be made.

For the cLi-greases:− Being insoluble in water, silica and calcium carbonate have no significant impact on the physicochemical properties of cLi-lubricant when they are introduced before the heat treatment stage, while their addition after thermomechanical dispersion impinges the process of lithium complex soap formation.− Partially water-soluble calcium borate reacts with the components of the thickener and is recommended for addition only after the heat treatment stage.− The addition of halloysite and two-component NFA based on halloysite is characterized by the same impact on the properties of cLi-lubricant.− All of the used NFAs improved the anti-scuffing characteristics of the cLi-lubricant, and the greatest effect is reached when using silica and calcium borate.− The anti-wear properties of cLi-lubricant are decreased when greases are modified with the NFA based on montmorillonite and halloysite, and improved when using silicon dioxide, calcium carbonate, and calcium borate.

For PU-greases:− The addition of NFA in PU-lubricant before the heat treatment stage leads to an increase of yield stress and improved colloidal stability and dropping point, but the effective viscosity decreases. Probably, NFAs participate in forming the structure of PU-thickener as active centers, which intensify the reactions between the components of the dispersion phase; taking into consideration the results of the research presented in [35], one can assume the possibility of obtaining PU-lubricants with desired rheological properties by introducing the NFA and using base oils of different types.− All the applied NFAs improve the anti-scuffing characteristics of PU-lubricant, particularly when calcium carbonate and the two-component NFA based on halloysite are used.− Anti-wear properties of PU-grease are improved when silica and halloysite are added.

For PP-greases:− There is a similar dependence of the PP-lubricant characteristics and their composition on the addition of nanoscale particles of silica, halloysite, and the two-component additive based on halloysite.− A similar efficiency level is shown by the NFAs of calcium carbonate and borate.− The use of calcium carbonate leads to a significant improvement of anti-scuffing properties when it is introduced after the heat treatment stage.− A remarkable decrease of wear scar diameter can be noticed when using the two-component NFA based on halloysite.− NFA addition does not negatively affect the process of the structure formation of PP-lubricants, unlike lithium stearate, which, when added even in low concentrations, leads to drastic worsening of the colloidal-mechanical characteristics of polypropylene-based lubricants [36]. 

Based on the results obtained, the nanosized particles of silica, calcium carbonate, and calcium borate were selected as the objects of study for the impact of NFA concentration on the cLi-greases’ and PU-greases’ properties.

### 3.3. Investigation of the Impact of the Selected NFA Concentration on the Characteristics of the cLi-Grease and PU-Grease

At this stage, we investigated the influence of concentration of nanosized particles of silica, calcium carbonate, and calcium borate on the physicochemical and anti-wear and anti-scuffing properties of cLi- and PU-greases. The applied concentrations of NFAs were in the range of 1–3 wt.%. The addition of the NFA to cLi-grease was carried out in the following order: silica and calcium carbonate were introduced before the heat treatment and calcium borate was introduced after. All the NFAs were introduced into the PU-lubricant before the thermomechanical dispersion. The results of the study are shown in Appendix A, respectively.

#### 3.3.1. Effect of Silica Concentration

An increase in the concentration of aerosil in the composition of cLi-grease entailrf an increase in the yield stress and the effective viscosity and improved colloidal stability of the lubricant. However, the anti-scuffing properties stayed the same, and the anti-wear characteristics were slightly degraded. This is explained by the interaction of hydroxyl groups contained in the surface layer of NFA with the oxygen of the acyl groups of complex lithium soap molecules, because of which the specific surface area of the thickener was increased, along with its strength and sorption activity towards the dispersion medium. The increase in metal wear when increasing the concentration of aerosil is explained by the abrasive properties of the latter.

Unlike cLi-grease, the increase of the silica content in the composition of PU-grease led to a decrease in the yield stress and an increase in the effective viscosity, which is most likely due to the mutual orientation of the NFA hydroxyls and the oxygen of the cyanate groups, which led to a decrease in the molecular weight of the polyurea molecules formed with a simultaneous increase in the specific surface of the thickener. Also, the worsening of anti-wear and anti-scuffing characteristics was noted with the increase of the concentration of aerosil in the composition of PU-grease.

#### 3.3.2. Effect of Calcium Carbonate Concentration

The effect of the increase of the content of calcium carbonate in cLi-grease consisted of the growth of its yield stress and effective viscosity and improved colloidal stability due to the inclusion of calcium carbonate nanoparticles in the three-dimensional structural framework of the thickener in the process of its crystallization. A slight improvement in the anti-wear and anti-scuffing properties of modified greases was also observed, similar to the results of studies on the effect of this NFA on simple lithium greases presented in [18].

In a PU-grease, when increasing the concentration of calcium carbonate nanoparticles, the decrease of the yield stress and colloidal stability occurred with the simultaneous increase of the effective viscosity, which can be explained by a decrease in the length and an increase in the number of polyurea thickener chains formed due to the increase of centers of the active reaction; nanoscale particles of calcium carbonate with a developed surface act as such centers. Along with a slight improvement of anti-wear characteristics, the anti-scuffing properties of PU-lubricants improved drastically.

#### 3.3.3. Effect of Calcium Borate Concentration

Unlike silicon dioxide and calcium carbonate, the increased amount of nanosized particles of calcium borate added into cLi-grease led to a significant reduction of its yield stress, poor colloid stability, and reduced effective viscosity that may have been caused by the increase in the number of three-component molecules of complex soap which formed a less dense structure of the thickener fibers with less thickening ability. The anti-wear and anti-scuffing characteristics of the cLi-lubricant remained almost unchanged, which indicates different degrees of this NFA’s influence on the properties of simple and complex lithium greases [20].

Changing the concentration of calcium borate nanoparticles in the composition of PU-grease from 1 to 3 wt.% almost did not affect its main characteristics except for a slight improvement of its anti-wear and anti-scuffing properties.

Thus, as a result of the study of the characteristics of cLi-grease and PU-grease depending on the concentration of nanosized particles of silicon dioxide, calcium carbonate, and borate, a number of conclusions can be made.

For cLi-grease:− An increase in the concentration of silica and calcium carbonate NFA in the studied range leads to the hardening of the structure of the cLi-grease, while the increase of the concentration of nanosized particles of calcium borate softens the structure of the dispersed phase of cLi-grease.− A maximal improvement of anti-scuffing characteristics of the cLi-lubricant is reached when 1% of NFA is introduced; a further increase of the concentration of the tested nanoscale additives does not bring significant changes to the anti-wear and anti-scuffing properties.− Among the considered NFAs, calcium borate nanoparticles have the highest efficiency as an additive which improves the anti-wear and anti-scuffing characteristics of cLi-lubricants.

For PU-grease:− The increase in the concentration of silicon dioxide and calcium carbonate from 1 to 3 wt.% leads to a decrease in yield stress and the deterioration of colloidal stability. − With an increasing concentration of all the investigated NFAs, the effective viscosity of cLi- and PU-greases increases. − The worst dynamics of changes in the anti-wear and anti-scuffing properties of the PU-lubricant is observed when the concentration of silicon dioxide nanosized particles is increased.− The maximal improvement of the anti-scuffing characteristics of PU-grease can be reached with the increase of the calcium carbonate NFA concentration, and for anti-wear characteristics with the increase of the calcium borate. 

## 4. Conclusions

The analysis of the literature revealed that NFA application is a promising direction for the improvement of the anti-wear, anti-scuffing, and rheological characteristics of greases. The scale of the effect depends on factors such as the size and nature of the nanoparticles, the composition of the greases, and the concentration and the method of nanosized particles addition into the structure of the lubricant.

When modifying the cLi-greases, it was found that the addition of NFA did not have a clear positive effect on the physicochemical characteristics of the lubricant, but it led to a significant change in the anti-scuffing and anti-wear properties. The maximal improvement of the anti-wear and anti-scuffing characteristics of cLi-greases was reached when using silica and calcium borate.

The addition method and the concentration of NFA had a serious influence on the characteristics of cLi-greases. The addition of silicon dioxide and calcium carbonate NFA after the heat treatment stage led to worsening characteristics of the plastic material, and the increase of their concentration from 1 to 3 wt.% formed a harder structure of Li-grease. On the contrary, the addition of calcium borate NFA is recommended after the thermomechanical dispersion. The increase of the concentration of calcium borate nanosized particles had a negative effect on the yield stress of the grease. When NFA was introduced at 1 wt.%, the maximal anti-scuffing effect of cLi-grease was reached, whereas the increase of its concentration up to 3 wt.% did not bring a significant change in the anti-wear and anti-scuffing performance.

The addition of the NFA before the thermomechanical dispersion in the production process of PU-lubricant had a significant impact on its end physicochemical characteristics. The maximal improvement of the anti-scuffing properties of the PU-lubricant was reached when using calcium carbonate and the two-component NFA based on halloysite, and the maximal improvement for the anti-wear properties was reached when adding silicon dioxide and halloysite.

When the concentrations of silicon dioxide and calcium carbonate were increased from 1 to 3 wt.%, there was a decrease in the yield stress of the structural frame of the PU-lubricant and its colloidal stability was worse. The increase of the concentration of calcium carbonate and borate nanoparticles in the studied range led to a significant improvement of the anti-scuffing and anti-wear characteristics of the PU grease, respectively.

When modifying the PP-grease, the improvement of its anti-scuffing properties was noted when the calcium carbonate NFA was introduced after the heat treatment stage, while the use of the two-component NFA based on halloysite had a positive effect on the anti-wear characteristics.

Thus, the data obtained during the experimental studies allowed us to determine the dependences of the main physicochemical and anti-wear and anti-scuffing characteristics of LTG with various thickeners on the type, method of addition, and concentration of NFA.

## Figures and Tables

**Figure 1 polymers-13-03749-f001:**
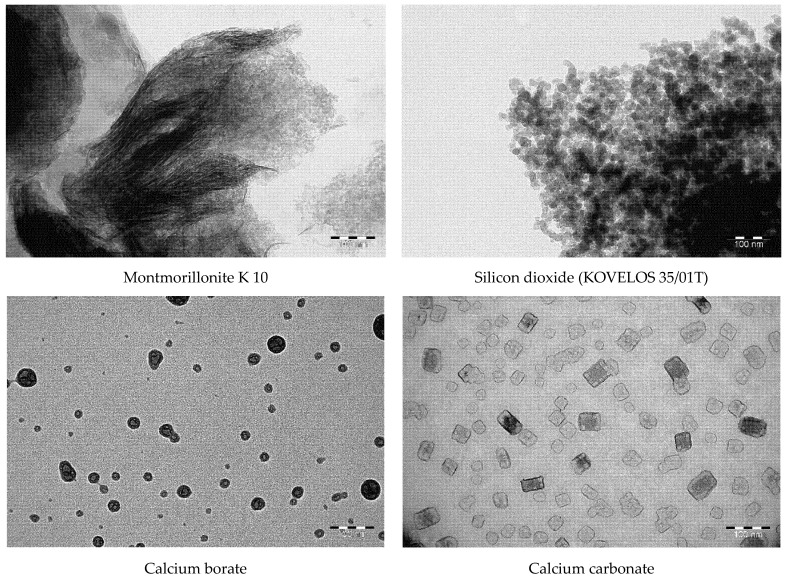
Morphology and microstructure of the applied nanoparticles.

**Table 1 polymers-13-03749-t001:** The characteristics of NFAs.

NFA	Specific Surface Area, m^2^/g
Montmorillonite K 10	245.0
Silica (Kovelos 35/01T)	350.0
Calcium borate	207.6
Calcium carbonate	183.1
Halloysite	71.8
Molybdenum disulfide incorporated into halloysite tubes	46.0

**Table 2 polymers-13-03749-t002:** The characteristics of the base oil S-9.

Parameter	Kinematic Viscosity, cSt		Viscosity Index	Pour Point, °C	Open-Cup Flash-point, °C	CCS, Minus 35 °C, mPa∙s	MRV, Minus 40 °C, mPa s	NOACK Test,%	Group composition BashNII test, hydrocarbon content,% weight: - paraffin-naphthene - light aromatic - medium aromatic - heavy aromatic
	at 40 °C	at 100 °C							
Method	ASTM D 445		ASTM D 2270	ASTM D 97	ASTM D 92	ASTM D 5293	ASTM D 4684	ASTM D 5800	BashNII Test
Oil S-9	10.88	2.74	86	minus 50	176	1076	2005	62.97	91.3 4.4 1.5 2.8

## Data Availability

The data in this paper are available upon request from the corresponding author.

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
