# Peer review of "Nanoscale Functional Additives Application in the Low Temperature Greases"

_polymers, 2021, doi:10.3390/polym13213749_

Round 1

Reviewer 1 Report

In the focus of the draft paper are the impacts of six different types of nanoscale functional additives on lubricating greases. Therefore, the authors had a look at the nanosized particles and mixed them in different ways into three different kinds of greases for further studies. With these 36 variants they measured important grease properties as well as the scuffing load and wear resistance in a four-ball apparatus. In a second step, the authors chose the three combinations with the biggest improvements compared to the plain grease and repeat the investigations under variation of the nanoparticle concentration. At the end, they classify the different influences on the greases and their lubricating properties.

Overall, the paper makes a good impression on me. The investigations are described in detail from the mixing procedure of the greases to the results of the four-ball apparatus. The experimental program is appropriate for the desired results. The paper is structured in a comprehensible way. The observations are classified in the state of knowledge and the results seem plausible.

My only points of criticism are:

- The English needs to be revised. There are several language mistakes that result in irritations. For example, the word “introduction” is used instead of “addition” (of nanoscale particles), “the next nanoparticles” instead of “the following nanoparticles”. I have not understood, what “low tribological characteristics” are. Instead of “depletion of the metal” something like “wear”, “damage” or “fatigue” can be used and I would prefer “concentration” to “percentage of the involving” (all examples on p.1). The authors should have a closer look at the language in the whole paper.

- I think the authors have mixed the codes of the grease thickeners. They are introduced as “cLi”, “PU” and “PP” (p.5). In the further discussion they use “cLi”, “PP” and “PM”.

- I do not understand why the authors added the nanoparticles before and after the heat treatment. What was their intension? A small paragraph in the introduction about possible differences between these variants and the author’s expectations would be helpful.

Finally, one proposal to the authors: It would be easier to understand the results and follow the discussion, if they are not only presented in tables in the appendix, but in the text at the respective position. Maybe it would help much more as a diagram or graphic, if possible.

All in all, I enjoyed reading the paper. Thus, I would suggest “minor revisions” due to my points mentioned above.

Author Response

Response to Reviewer 1 Comments

Point 1: The English needs to be revised. There are several language mistakes that result in irritations. For example, the word “introduction” is used instead of “addition” (of nanoscale particles), “the next nanoparticles” instead of “the following nanoparticles”. I have not understood, what “low tribological characteristics” are. Instead of “depletion of the metal” something like “wear”, “damage” or “fatigue” can be used and I would prefer “concentration” to “percentage of the involving” (all examples on p.1). The authors should have a closer look at the language in the whole paper.

Response 1: Dear Reviewer, thank you very much for your valuable comments. We made necessary changes to meet your suggestions. Mistakes were corrected in the text.

Point 2: I think the authors have mixed the codes of the grease thickeners. They are introduced as “cLi”, “PU” and “PP” (p.5). In the further discussion they use “cLi”, “PP” and “PM”.

Response 2: The codes of the grease thickeners were change.

Point 3: I do not understand why the authors added the nanoparticles before and after the heat treatment. What was their intension? A small paragraph in the introduction about possible differences between these variants and the author’s expectations would be helpful.

Response 3: Additional comments were added in the introduction. The moment of particle addition significantly affects the process of structure formation of the lubricant and its physicochemical and operational characteristics.

Point 4: Finally, one proposal to the authors: It would be easier to understand the results and follow the discussion, if they are not only presented in tables in the appendix, but in the text at the respective position. Maybe it would help much more as a diagram or graphic, if possible.

Response 4: Several changes were made to improve comprehension and clarity of the text.

Reviewer 2 Report

The authors have performed a study of the influence of different type of nanoadditives on the tribological performance of low temperature greases. There are several reasons why I suggest this article to be rejected:

Minor comments.

  • The materials are not correctly described (company, reference and batch).
  • The presentation of the data in Table 2 can be easily improved.

Major comments.

In the supporting information, we can find some of the data that the authors claim that they have done. However, I miss the following figures:

  1. The wear scars have been measured, but I would like to see the perfilometry and the SEM pictures. Not only the values of wear are important, but the mechanism of the wear and debris are also relevant.
  2. Where are the coefficients of friction?
  3. The authors talk about viscosity, but the have not checked if the greases are Newtonian.
  4. How are the thermal stability affected by the addition of NFA?
  5. I also think that the particle debris should be analyzed.
  6. I cannot see a proper description of the tribological tests (load, speed, contacts, etc.).

Author Response

Response to Reviewer 2 Comments

Point 1: In the supporting information, we can find some of the data that the authors claim that they have done. However, I miss the following figures:

  1. The wear scars have been measured, but I would like to see the perfilometry and the SEM pictures. Not only the values of wear are important, but the mechanism of the wear and debris are also relevant.
  1. Where are the coefficients of friction?

Response 1:

Respected Reviewer, thank you very much for your valuable comments and support.

Authors investigated such main physicochemical characteristics as tensile strength, colloidal stability, dropping point, effective viscosity at minus 50 ° and basic tribological characteristics. The test methods proposed by the reviewer (study of the wear mechanism, friction coefficients, particle debris) are important, but highly specialized methods for assessing the tribological properties of greases and have not been studied within the framework of this work.

Point 2: The authors talk about viscosity, but they have not checked if the greases are Newtonian.

Response 2: Lubricants are non-Newtonian fluids. Within the framework of this work, the effective viscosity was estimated at a temperature of minus 50 ° C and an average strain rate gradient of 10 s-1 according to GOST 7163.

Point 3: How are the thermal stability affected by the addition of NFA?

Response 3: It is known that NFA in greases in the studied concentrations does not have a noticeable effect on the thermal stability of greases.

Point 4: I also think that the particle debris should be analyzed.

Response 4: The research was made with the assumption that impurities do not affect the properties. The study of the effect of impurities should be carried out outside the framework of this study.

Point 5: I cannot see a proper description of the tribological tests (load, speed, contacts, etc.).

Response 5: Measurement of tribological characteristics was carried out in accordance with GOST 9490. Liquid lubricating and plastic materials. Method of test for lubricating properties on four-ball machine.

Reviewer 3 Report

This paper investigated the performance of various nanoscale functional additives such as calcium borate, calcium carbonate, halloysite, and MoS2 on halloysite. Overall, the manuscript is well-organized and establishing tribological mechanism behind nanoadditives. However, some experimental results are omitted, and theoretical approaches are missing. Reviewer recommends major revision for this manuscript to be improved. Detailed comments for the revision are followed below.

  • The abstract is too broad and with too little details. What are the findings of this research? What is the relevance and novelty of this study? Strong statements about this paper are needed to strengthen the novelty of this paper.
  • The author highlights the lubricated nanostructured functional additives (NFA) in greases. However, there is no specific explanation why the author chose NFA to improve the tribological performance of the lubricant. References are partly missing and the actual state of knowledge about the role of nanoadditives in lubricant is completely missing. Giving the full-level of description to the role and characteristics of these lubricant and additive would improve this paper.
  • Please provide scheme to explain research outline and the experimental process of the study. It helps readers to understand overall story of this study.
  • One critical problem of this paper is that there is lack of materials characterization. Only SEM images were investigated to study morphological structure of each nanoadditive. AFM or TEM images need to highlight the feature of each nanoadditive. Moreover, chemical analysis such as XPS or EDX needs to be performed for characterization the fundamental chemical structure of the nanoadditives.
  • Scale bars for SEM images are invisible.

  • Authors mentioned that friction and wear were critical factors to analyze the performance of nanoadditives in lubricant. However, there are lack of data analysis which relate with friction and wear. What about friction coefficient data? What about wear track data? Specifically, giving the line profile over wear track would help readers grasping the morphological degree of wear performance with including nanoadditives.
  • In conclusion part, future works and contributive value to tribological field from the results of this manuscript would be helpful to emphasize the novelty of this manuscript.

Author Response

Response to Reviewer 3 Comments

Point 1: This paper investigated the performance of various nanoscale functional additives such as calcium borate, calcium carbonate, halloysite, and MoS2 on halloysite. Overall, the manuscript is well-organized and establishing tribological mechanism behind nanoadditives. However, some experimental results are omitted, and theoretical approaches are missing. Reviewer recommends major revision for this manuscript to be improved. Detailed comments for the revision are followed below.

The abstract is too broad and with too little details. What are the findings of this research? What is the relevance and novelty of this study? Strong statements about this paper are needed to strengthen the novelty of this paper.

Response 1:

Respected Reviewer, thank you very much for your valuable comments and support. The abstract was improved according to your comments.

Point 2: The author highlights the lubricated nanostructured functional additives (NFA) in greases. However, there is no specific explanation why the author chose NFA to improve the tribological performance of the lubricant. References are partly missing and the actual state of knowledge about the role of nanoadditives in lubricant is completely missing. Giving the full-level of description to the role and characteristics of these lubricant and additive would improve this paper.

Response 2: The mechanism of action of NFA is briefly described in the introduction. Their advantages over traditional EP additives are also presented. Also, in the introduction, references are given to [4,5,9,13-16,20-35], which provide comprehensive information on the role of nanoadditives in lubricants.

Point 3: Please provide scheme to explain research outline and the experimental process of the study. It helps readers to understand overall story of this study.

Response 3: Initially, 6 NFA and 3 greases were chosen. NFA was introduced into the test lubricants equal concentration before and after the thermo-mechanical dispersion stage. 36 samples were obtained. The most advantageous in terms of physicochemical characteristics were selected and further investigated. Three selected NFA were added at three different concentrations to complex lithium and polyurea greases. All obtained samples were examined in accordance with the specified GOST methods.

Point 4: One critical problem of this paper is that there is lack of materials characterization. Only SEM images were investigated to study morphological structure of each nanoadditive. AFM or TEM images need to highlight the feature of each nanoadditive. Moreover, chemical analysis such as XPS or EDX needs to be performed for characterization the fundamental chemical structure of the nanoadditives.

Response 4: The study of AFM, TEM and NFA chemical analysis required to study the use of greases is an important area of research and is currently underway.

Point 5: Scale bars for SEM images are invisible.

Response 5: Scale bars for SEM images is 100 nm.

Point 6: Authors mentioned that friction and wear were critical factors to analyze the performance of nanoadditives in lubricant. However, there are lack of data analysis which relate with friction and wear. What about friction coefficient data? What about wear track data? Specifically, giving the line profile over wear track would help readers grasping the morphological degree of wear performance with including nanoadditives.

Response 6: Authors investigated such main physicochemical characteristics as tensile strength, colloidal stability, dropping point, effective viscosity at minus 50 ° and basic tribological characteristics. The test methods proposed by the reviewer (study of the wear mechanism, friction coefficients, particle debris) are important, but highly specialized methods for assessing the tribological properties of greases and have not been studied within the framework of this work.

Round 2

Reviewer 2 Report

The authors claim that the study of wear mechanism, friction coefficient and particle debris are out of the scope of this manuscript. However, in the conclusions they wrote the following paragraph:

"When modifying the cLi-greases, it was found that the addition of NFA does not have clear positive effect on the physicochemical characteristics of the lubricant, but it leads to the significant change in the antiscuffing and antiwear properties. Maximal improvement of tribological characteristics of cLi-greases was reached when using silica and calcium borate."

And

"Thus, the data obtained during the experimental studies allowed us to determine the dependences of the main physicochemical and tribological characteristics of LTG with various of thickeners on the type, method of addition and concentration of NFA."

The problem is that they are claiming that the addition of NFA improves the tribological performance of greases, but they have not measured the basic tribological properties.

The conditions of the testing are just explained with a standard. They should provide the load, speed, etc... and also the equipment.

The answer " It is known that NFA in greases in the studied concentrations does not have a noticeable effect on the thermal stability of greases.", please provide the literature.

I still recommend this article to be rejected since it does not fulfill the quality standards of Polymers.

Author Response

Comments and Suggestions for Authors

Point 1: The authors claim that the study of wear mechanism, friction coefficient and particle debris are out of the scope of this manuscript. However, in the conclusions they wrote the following paragraph:

"When modifying the cLi-greases, it was found that the addition of NFA does not have clear positive effect on the physicochemical characteristics of the lubricant, but it leads to the significant change in the antiscuffing and antiwear properties. Maximal improvement of tribological characteristics of cLi-greases was reached when using silica and calcium borate."

And

"Thus, the data obtained during the experimental studies allowed us to determine the dependences of the main physicochemical and tribological characteristics of LTG with various of thickeners on the type, method of addition and concentration of NFA."

The problem is that they are claiming that the addition of NFA improves the tribological performance of greases, but they have not measured the basic tribological properties.

Response 1: Currently for lubricating properties' evaluation producers of low-temperature greases use wear scar and welding load indicators characterizing antiwear and antiscuffing properties respectively.

Point 2: The conditions of the testing are just explained with a standard. They should provide the load, speed, etc… and also the equipment.

Response 2: The test conditions can be found in GOST 9490-75: "Liquid lubricating and plastic materials: Method of test for lubricating characteristics on a four-ball machine."

Point 3: The answer " It is known that NFA in greases in the studied concentrations does not have a noticeable effect on the thermal stability of greases.", please provide the literature.

Response 3: The information proved this fact you can find in Fuchs I. G. Greases. M.: Chemistry, 1972. 158 p.

Reviewer 3 Report

The authors did not make adequate revisions about most of reviewer's major points. Reviewer do not recommend publish without improving the manuscript.

Author Response

Point 1: This paper investigated the performance of various nanoscale functional additives such as calcium borate, calcium carbonate, halloysite, and MoS2 on halloysite. Overall, the manuscript is well-organized and establishing tribological mechanism behind nanoadditives. However, some experimental results are omitted, and theoretical approaches are missing. Reviewer recommends major revision for this manuscript to be improved. Detailed comments for the revision are followed below.

The abstract is too broad and with too little details. What are the findings of this research? What is the relevance and novelty of this study? Strong statements about this paper are needed to strengthen the novelty of this paper.

Response 1:

Relevance: Due to the fact that the application of AW and EP additives in low-temperature greases may lead to worse high-temperature and anti-corrosion characteristics as well as to additional load on the environment due to the content of aggressive components, in this paper, the possibility of replacing these additives with NFA, which do not have these disadvantages, was investigated.

Novelty: Changes in the antiwear and antiscuffing properties of low-temperature greases after the addition of NFA into them were evaluated. The influence of the size and nature of the nanoparticles, the composition of greases, the concentration and the method of nanosized particles addition into the structure of the lubricant on antiwear and antiscuffing properties is studied.

Results: The possibility of production low-temperature greases, in which AW and EP additives are replaced with NFA, with excellent lubricating properties, has been established.

The abstract was improved based on Reviewer’s comments.

Point 2: The author highlights the lubricated nanostructured functional additives (NFA) in greases. However, there is no specific explanation why the author chose NFA to improve the tribological performance of the lubricant. References are partly missing and the actual state of knowledge about the role of nanoadditives in lubricant is completely missing. Giving the full-level of description to the role and characteristics of these lubricant and additive would improve this paper.

Response 2: The mechanism of action of NFA is briefly described in the introduction. Their advantages over traditional EP additives are also presented. Also, in the introduction, references are given to [4,5,9,13-16,20-35], which provide comprehensive information on the role of nanoadditives in lubricants.

Point 3: Please provide scheme to explain research outline and the experimental process of the study. It helps readers to understand overall story of this study.

Response 3: Initially, 6 NFA and 3 greases were chosen. NFA was introduced into the test lubricants equal concentration before and after the thermo-mechanical dispersion stage. 36 samples were obtained. The most advantageous in terms of physicochemical characteristics were selected and further investigated. Three selected NFA were added at three different concentrations to complex lithium and polyurea greases. All obtained samples were examined in accordance with the specified GOST methods.

Point 4: One critical problem of this paper is that there is lack of materials characterization. Only SEM images were investigated to study morphological structure of each nanoadditive. AFM or TEM images need to highlight the feature of each nanoadditive. Moreover, chemical analysis such as XPS or EDX needs to be performed for characterization the fundamental chemical structure of the nanoadditives.

Response 4: The study of AFM, TEM and NFA chemical analysis required to study the use of greases is an important area of research and is currently underway.

Point 5: Scale bars for SEM images are invisible.

Response 5: Scale bars for SEM images is 100 nm.

Point 6: Authors mentioned that friction and wear were critical factors to analyze the performance of nanoadditives in lubricant. However, there are lack of data analysis which relate with friction and wear. What about friction coefficient data? What about wear track data? Specifically, giving the line profile over wear track would help readers grasping the morphological degree of wear performance with including nanoadditives.

Response 6: Currently for lubricating properties' evaluation producers of low-temperature greases use wear scar and welding load indicators characterizing antiwear and antiscuffing properties respectively. In this regard, these indicators were used to evaluate the lubricating properties in the study.

Round 3

Reviewer 2 Report

With these last changes, the manuscript can be accepted for publication.

Reviewer 3 Report

The authors still did not make adequate revisions about most of reviewer's major points. ( especially point 3, 4, 6)

Author Response

Dear Reviewer, thank you very much for your valuable comments. Please find below the revised response to your questions. Authors present their point of view more clearly and include additional information in the response.

Point 1: This paper investigated the performance of various nanoscale functional additives such as calcium borate, calcium carbonate, halloysite, and MoS2 on halloysite. Overall, the manuscript is well-organized and establishing tribological mechanism behind nanoadditives. However, some experimental results are omitted, and theoretical approaches are missing. Reviewer recommends major revision for this manuscript to be improved. Detailed comments for the revision are followed below.

The abstract is too broad and with too little details. What are the findings of this research? What is the relevance and novelty of this study? Strong statements about this paper are needed to strengthen the novelty of this paper.

Response 1:

            Currently, many indicators are used to assess the possibility of using low-temperature lubricants in various equipment. The work used those indicators that characterize the performance properties of the lubricant, as well as those on which the introduction of nanostructured functional additives (NFA) could have the greatest impact - Yield stress at 50 °Ð¡, Pa; Colloidal stability, % of oil emitted; Dropping point, °Ð¡; Effective viscosity at minus 50 ºÐ¡, Pa·s; Lubricating properties on a four-ball friction machine under the temperature of (20 ± 5) °Ð¡: welding load, kgs, the wear scar, mm. In the work, such indicators as penetration were omitted, since this indicator characterizes the degree of readiness of the product during the production process and does not reflect operational properties in any way.

Relevance: Due to the fact that the application of AW and EP additives in low-temperature greases may lead to worse high-temperature and anti-corrosion characteristics as well as to additional load on the environment due to the content of aggressive components, in this paper, the possibility of replacing these additives with NFA, which do not have these disadvantages, was investigated.

Novelty: Changes in the antiwear and antiscuffing properties of low-temperature greases after the addition of NFA into them were evaluated. The influence of the size and nature of the nanoparticles, the composition of greases, the concentration and the method of nanosized particles addition into the structure of the lubricant on antiwear and antiscuffing properties is studied.

Results: The possibility of production low-temperature greases, in which AW and EP additives are replaced with NFA, with excellent lubricating properties, has been established.

The main points of novelty and results are included in the abstract.

Point 2: The author highlights the lubricated nanostructured functional additives (NFA) in greases. However, there is no specific explanation why the author chose NFA to improve the tribological performance of the lubricant. References are partly missing and the actual state of knowledge about the role of nanoadditives in lubricant is completely missing. Giving the full-level of description to the role and characteristics of these lubricant and additive would improve this paper.

Response 2: Nanostructured functional additives (NFA) of various composition and origin are introduced into lubricants, which, as a rule, have a high lubricity. The introduction of nanostructured functional additives (NFA) into lubricants pursues a variety of goals, namely: improving lubricity, increasing sealing properties and protective ability, increasing the strength of the lubricant, reducing its extrusion from friction units, increasing heat resistance, reducing the coefficient of friction. In some cases, nanostructured functional additives (NFA) can be introduced for dilution in order to save expensive or scarce lubricants, provided there is no change in product quality. The properties of the filled lubricants are determined by the nature of the interaction of the nanostructured functional additives (NFA) with the other components of the lubricant. Therefore, in addition to the composition of the lubricant, the nature, concentration, particle size of the nanostructured functional additives (NFA) and the method of its introduction are important.

The mechanism of action of NFA is briefly described in the introduction. Their advantages over traditional EP additives are also presented. Also, in the introduction, references are given to [4,5,9,13-16,20-35], which provide comprehensive information on the role of nanoadditives in lubricants.

Probably, the authors did not include complete information on the role of nanoadditives in lubricants with the corresponding references. Dear Reviewer, could you please advise us on which references should be included in the paper?

Point 3: Please provide scheme to explain research outline and the experimental process of the study. It helps readers to understand overall story of this study.

Response 3: Initially, 6 NFA and 3 greases were chosen. NFA was introduced into the test lubricants equal concentration before and after the thermo-mechanical dispersion stage. 36 samples were obtained. The comparison was carried out according to the following indicators: Yield stress at 50 °Ð¡, Pa; Colloidal stability, % of oil emitted; Dropping point, °Ð¡; Effective viscosity at minus 50 ºÐ¡, Pa·s; Lubricating properties on a four-ball friction machine under the temperature of (20 ± 5) °Ð¡: welding load, kgs, the wear scar, mm. The most advantageous were selected and further investigated. Three selected NFA were added at three different concentrations to complex lithium and polyurea greases. All obtained samples were examined in accordance with the specified GOST methods.

Point 4: One critical problem of this paper is that there is lack of materials characterization. Only SEM images were investigated to study morphological structure of each nanoadditive. AFM or TEM images need to highlight the feature of each nanoadditive. Moreover, chemical analysis such as XPS or EDX needs to be performed for characterization the fundamental chemical structure of the nanoadditives.

Response 4: The plan of this work implied an assessment of the tribological characteristics - antiwear and antiscuffing properties, of various combinations of components: 3 greases – complex lithium, polyurea and polymer, and 6 nanostructured functional additives (NFA) - montmorillonite K 10, silica, calcium carbonate and borate, halloysite and molybdenum disulfide incorporated in halloysite tubes, in connection with which the SEM method was used. There is no doubt that AFM and TEM are important for highlighting the feature of each nanoadditive, and the XPS and EDX methods are informative for characterization the fundamental chemical structure of the nanoadditives. But the amount of work required to carry out these analyses will be the basis for a separate scientific work.

Point 5: Scale bars for SEM images are invisible.

Response 5: Scale bars for SEM images is 100 nm.

Point 6: Authors mentioned that friction and wear were critical factors to analyze the performance of nanoadditives in lubricant. However, there are lack of data analysis which relate with friction and wear. What about friction coefficient data? What about wear track data? Specifically, giving the line profile over wear track would help readers grasping the morphological degree of wear performance with including nanoadditives.

Response 6: The lubricating properties of greases include antifriction properties, antiwear properties, antiscuffing properties, antipitting properties and anti-«stick-slip» properties.   Antifriction properties are properties of a lubricant that characterize the ability to reduce the coefficient of friction of solids in a certain range of temperatures, pressures, relative motion speeds and combinations of rubbing materials.

Antiwear properties are properties of a lubricant that characterize the ability to reduce intensive wear to specified parameters at nominal load on the friction unit.

Antiscuffing properties are properties of a lubricant to prevent adhesive setting in conditions of short-term supernominal loads.

Antipitting properties are properties of a lubricant that characterize the ability to prevent fatigue staining (pitting) of the friction surface under cyclic loads.

Anti-«stick-slip» properties are properties of a lubricant that characterize the ability to stabilize the coefficient of friction.

Currently for lubricating properties' evaluation producers of low-temperature greases use wear scar and welding load indicators characterizing antiwear and antiscuffing properties respectively. In this regard, these indicators were used to evaluate the lubricating properties in the study.